# Outcome Predictor Differences in Infratentorial and Supratentorial Ischemic Stroke

**DOI:** 10.3390/life15040633

**Published:** 2025-04-10

**Authors:** Manuel Bolognese, Mareike Österreich, Martin Müller, Alexander von Hessling, Grzegorz Marek Karwacki, Lehel-Barna Lakatos

**Affiliations:** 1Department of Neurology and Neurorehabilitation, Lucerne Cantonal Hospital, 6000 Lucerne, Switzerland; manuel.bolognese@luks.ch (M.B.); mareike.oesterreich@luks.ch (M.Ö.); lehel-barna.lakatos@luks.ch (L.-B.L.); 2Department of Radiology, Section Neuroradiology, Lucerne Cantonal Hospital, 6000 Lucerne, Switzerland; alexander.vonhessling@luks.ch (A.v.H.); grzegorz.karwacki@luks.ch (G.M.K.)

**Keywords:** biomarkers, carotid artery, chronic kidney disease, stroke etiology, stroke, Troponin, vertebrobasilar artery

## Abstract

Acute ischemic stroke outcomes depend on various factors. We investigated whether the outcome-relevant factor (ORF) profiles differ between different vascular territories and different therapeutic strategies. In this retrospective study, we analyzed 410 comprehensive stroke center patients [median age of 70 years (IQR 57–80), 125 women (30%)] by analyzing five groups: all patients, patients with infratentorial infarctions only (n = 80), all patients with supratentorial infarctions (n = 330), patients with supratentorial infarctions without (n = 269), and with mechanical thrombectomy (n = 61). Outcomes were classified with the modified Rankin scale as ≤2 (good) or >2 (poor) after three months. The patient group with infratentorial strokes was compared to the group of patients with supratentorial strokes using the Kruskal–Wallis test or chi-squared statistics. Within each of the five stroke groups, univariate logistic regression analysis was used to identify the ORF of a poor outcome; if more than one ORF was identified, all identified factors were included in one multinomial logistic regression analysis model. Compared to the patients with supratentorial strokes, the patients with infratentorial stroke exhibited a less severe neurological deficit at entry and lower rates of ischemic heart disease, thrombolytic intervention, and cardio-embolism but a higher rate of large vessel disease. After multinomial logistic regression analysis, a poor outcome in the infratentorial group was associated with atrial fibrillation [odds ratio (OR) 13.73 (95% confidence interval 1.05–181.89), *p* = 0.04], estimated glomerular filtration rate [OR 0.96 (0.91–0.99)], *p* = 0.02], and marginally with diabetes mellitus [OR 7.69 (0.96–62.63), *p* = 0.05]. In all three supratentorial stroke groups, the neurological deficit as scored by the National Institute of Health Stroke Scale [OR 1.32 (1.22–1.44), *p* < 0.0001] was predominantly associated with a poor outcome, accompanied by age only in the group of all supratentorial strokes [OR 1.04 (1.01–1.08), *p* = 0.01]. In this cohort of mild to moderate stroke patients, the ORFs differed between the supra- and infratentorial stroke populations.

## 1. Introduction

The introduction of intravenous thrombolysis (iv-lysis) and intra-arterial mechanical thrombectomy (MT) into the therapy of acute ischemic stroke improved the stroke outcome dramatically. In supratentorial stroke, MT improved the outcome on average by a change in the modified Rankin score (mRs) from 4 to 3 or, in other words, from a life depending on help in various aspects of daily living to a life independent from such help [1,2,3,4,5]. In vertebrobasilar artery (infratentorial) strokes, especially in basilar artery occlusions, mortality was decreased from as high as 80% to about 15%, and a favorable survival (mRs ≤3) can be observed in 40–50% of the patients receiving thrombolytic therapy [6,7,8,9]. Nevertheless, the classical outcome-relevant factors (ORFs) associated with a poor outcome such as age, sex, arterial hypertension, diabetes mellitus, smoking, lifestyle and resulting body mass index, chronic kidney disease, cardiac diseases (e.g., atrial fibrillation, cardio-embolism, and heart failure), race, and the initial clinical state of the patient are still major outcome determinants. In addition, new factors have emerged, including serological biomarkers such as high-sensitivity cardiac Troponin T (or I); the relevance of these factors compared to all other ORFs is, however, uncertain [10,11,12,13,14,15,16,17]. Studies investigating high-sensitivity cardiac Troponin T (hs-cTnT) specifically have generally reported the outcome relevance of hs-cTnT in cohorts that included both infra- and supratentorial strokes. Whether there are differences regarding the outcome-predictive ability of hs-cTnT with respect to different vascular territories has very rarely been addressed [18].

In general, future patient care is showing a trend toward more individualization. Knowledge of the ORFs and their relevance in a defined stroke population therefore seems increasingly important [18]. These subpopulations must be evaluated separately to determine the relevance of the ORFs in each subpopulation. In this retrospective study, we analyzed whether and, if so, which ORFs need special attention in a group of patients with infratentorial ischemic strokes compared to a group of patients with supratentorial ischemia.

## 2. Materials and Methods

We reviewed the patients included in the prospective clinical trial “Linearity and Non-Linearity of Cerebral Autoregulation” (registered at ClinicalTrials.gov NCT04611672), which investigated which factors influence dynamic cerebral autoregulation in stroke patients (n = 520, enrolled between 1 January 2020, and 31 April 2022). A formal statement of the local Ethics Committee of Northwest and Central Switzerland was not necessary for this retrospective analysis of routine data according to Swiss law. This study was conducted in accordance with the Declaration of Helsinki and adhered to good clinical practice standards. The corresponding author can provide all data for this study upon reasonable request.

### 2.1. Patients

The inclusion criteria for this present analysis were as follows: age of 18 years and a definitive supratentorial or infratentorial ischemic stroke after initial multimodal imaging with subsequent confirmation via diffusion-weighted imaging (DWI). Exclusion criteria included a final diagnosis of a stroke mimic (including the presence of a brain tumor), primary intracranial hemorrhage, transient ischemic attack, cerebral sinus or vein thrombosis as the cause of an acute neurological deficit, and pregnancy. Based on the AHA guidelines [19], we defined a priori all variables listed in Table 1 as ORFs and added hs-cTnT as a representative of the newer biomarkers. Because dynamic cerebral autoregulation is not an established outcome risk factor in infratentorial stroke, we did not include its analysis in this report.

### 2.2. Setting

The Lucerne Hospital is a large tertiary teaching hospital equipped with a comprehensive stroke center service. All patients presenting with stroke syndromes receive standardized care, which includes a focused clinical examination followed by multimodal cranial computed tomography (CT) using Siemens Force, Edge, or XCeed CT machines. This process involves native CT imaging, followed by CT perfusion (CTP) analyzed using Syngo.via (Siemens, Forchheim, Germany, version VB40D) and RapidAI Software (RAPIDAI, Golden, CO, USA, version 4.9.1.1) to estimate the infarct core and penumbra. A Tmax of >6 s indicates hypoperfusion (penumbra), while a relative decrease in cerebral blood flow (CBF) of 30% in comparison to the non-affected hemisphere indicates an infarct core. Additionally, CT angiography is performed to assess vascular status. If indicated, this is promptly followed by iv-lysis and/or MT. While recanalizing therapy in supratentorial infarctions follows established criteria [1,2,3,4,5], iv-lysis and MT in infratentorial strokes are discussed between neurologists and neuroradiologists based on the individual patient’s condition. All patients diagnosed with stroke syndromes are subsequently transferred to the stroke unit for intensive clinical monitoring. Monitoring includes the National Institute of Health Stroke Scale (NIHSS) [20] and mRs [21] assessments upon hospital entry, as well as daily assessments while in the stroke unit; the clinical scoring three months after the ischemic event is performed by personnel of the outpatient service team who are not involved in the scoring at hospital entry or at the stroke unit. Additionally, blood pressure, heart rate, body temperature, blood glucose level, and oxygen saturation are closely monitored. Within 72 h of hospitalization, ultrasound examination of the brain-supplying arteries, echocardiography, and brain magnetic resonance imaging [MRI, using DWI, T2-weighted, and susceptibility-weighted imaging sequences on either a Siemens Vida fit (3 Tesla), Siemens Aera (1.5 Tesla), or Philips Achieva (3 Tesla)], are performed. After all information is collected, stroke etiology is classified based on the Trial of Org 10172 in the Acute Stroke Treatment (TOAST) classification [22] into the following categories: cardio-embolism (CE), large vessel disease (LVD), lacunar (L), other determined causes (Others), or stroke of undetermined origin (Unknown), which indicates cases where no source or multiple sources of a stroke were identified. Blood samples are withdrawn as soon as the patients arrive at the hospital to estimate hs-cTnT levels (Roche Elecsys^®^ Immunoassay on the Roche Cobas 8000 system, Hoffmann-La Roche, Basel, Switzerland). The three-month outcome is assessed via our neurovascular outpatient service, which determines the mRs score and adherence to therapy (medication and physical therapy).

**Table 1 life-15-00633-t001:** Comparison of the baseline characteristics and the a priori-defined stroke outcome-relevant factors in the patients with infratentorial and supratentorial strokes.

	Infratentorial Stroke (n = 80)	Supratentorial Stroke (n = 330)	*p*
Baseline characteristic or predefined outcome risk factor			
Age (years)	70 (58–76)	71 (59–78)	0.25
Male/female n (%)	65 (82%):15 (18%)	230 (70%):110 (30%)	0.054
NIHSS score at entry	2 (IQR 1–4; range 0–37)	3 (IQR 1–7; range 0–27)	0.04
Body mass index (kg/m^2^)	25.9 (24.2–28.4)	25.7 (23.1–28.7)	0.33
Cardiac LVEF (%)	60 (55–64)	60 (55–64)	0.32
Arterial hypertension (%)	34 (42%)	131 (39%)	0.60
Diabetes mellitus (%)	24 (30%)	77 (23%)	0.24
Dyslipidemia (%)	62 (77%)	258 (78%)	0.88
Actual smoking (%)	18 (22.5%)	81 (24.5%)	0.66
Atrial fibrillation (%)	8 (10%)	57 (17%)	0.12
Ischemic heart disease (%)	9 (11%)	70 (21%)	0.04
Metabolic syndrome (%)	9 (11%)	42 (13%)	0.85
Intravenous thrombolysis (%)	18 (22.5%)	120 (36%)	0.01
Mechanical thrombectomy (%)	7 (8%)	61 (18%)	0.03
Hs-cTnT (ng/L) Median >14 (%)	10.5 (8–16) 23 (30%)	11 (6–20) 122 (37%)	0.77 0.19
eGFR (mL/min/1.73 m^2^) Median <60 (%)	83.5 (68–91) 15 (19%)	81 (66–90) 60 (19%)	0.11 0.87
TOAST classification of stroke (%) Cardio-embolism Large vessel disease Lacunar Others definite Unknown	15 (19%) 28 (35%) 18 (23%) 1 (1%) 18 (23%)	106 (32%) 53 (16%) 57 (18%) 9 (2%) 105 (31%)	0.02 0.0005 0.33 0.69 0.39
Outcome mRs 3 months Median ≤2:>2	1 (0–1) 70:10	1(0–2) 270:60	0.50 0.25

eGFR, estimated glomerular filtration rate; LVEF, left-ventricular ejection fraction; Hs-cTnT, high-sensitivity cardiac Troponin T; mRs, modified Rankin scale; NIHSS, National Institute of Health Stroke Scale; *p*, level of significance; and TOAST, TOAST classification according to Reference [22]. Hs-cTnT > 14 ng/L indicates a Troponin value above the 95% CI of the test’s upper limit range; eGFR ≤ 60 indicates chronic kidney disease. Values are given as numbers (percentages) or median (IQR). Only for the NIHSS score at entry, we additionally provide the range of the NIHSS to indicate overall stroke severity.

### 2.3. Statistics

The MATLAB Statistical Toolbox (Matlab version 2024a) was used for all data analyses. Categorical variables are reported as absolute numbers and percentages. We used either Fisher’s exact test or the chi-squared test to compare their distribution between groups. Continuous data that followed a normal distribution are expressed as mean ± standard deviation (SD), whereas non-normally distributed data are presented as median with interquartile range (IQR). Most continuous variables were not normally distributed; therefore, we used the non-parametric Kruskal–Wallis test for all between-group comparisons of continuous variables. The outcome after three months was classified as good (mRs ≤2) or poor (mRs >2). In the infratentorial and the supratentorial stroke groups, each of the predefined ORFs (Table 1) underwent a univariate logistic regression analysis to test their outcome relevance. Sample size calculations were performed using success probability for a binomial distribution at a power level of 0.8. In each stroke group, the ORFs demonstrating a significant relation to the outcome in the univariate analysis were then included in multinomial or multivariate logistic regression analysis models for further evaluation of their outcome relevance. In addition to logistic regression analysis, we performed a receiver operating characteristic curve (ROC) analysis with a calculation of the area under the curve (AUC) of each of the ORFs. A *p*-value of ≤0.05 was considered statistically significant.

## 3. Results

### 3.1. Patients

Out of 520 patients, 410 could be recruited for this study, 80 with infratentorial and 330 with supratentorial infarction (Figure 1). All our patients were of white Caucasian origin; thus, race was not included in our analysis. Their basic characteristics are reported in Table 1.

Some baseline characteristics/ORFs were significantly different between the two groups: on average, the patients in both groups were mild to moderately impaired at admission, with the supratentorial having a slightly higher NIHSS score by one point. In the supratentorial group, ischemic heart disease, recanalizing therapy with iv-lysis, and MT were more frequent. Regarding stroke etiology according to the TOAST classification, CE was more frequent in the supratentorial group, while LVD was found more often in the infratentorial group. The mRs at three months were not different between the two groups.

### 3.2. Outcome-Relevant Factors

Because we are interested in ORF profiles in different stroke populations, we report the ORFs first in all patients; second, we report the ORFs for each of the two subgroups, infratentorial and supratentorial; and third, we report the ORFs for the supratentorial stroke patients overall. We then divided these patients into those with and without MT. Because there were only seven MT procedures in the infratentorial stroke group, we did not conduct a comparison of whether or not MT was applied within this group.

#### 3.2.1. Relevance of the Predefined ORFs in the Outcome in the Total Stroke Population

The univariate logistic regression analysis (Table 2) revealed age, NIHSS score at entry, arterial hypertension, diabetes mellitus, atrial fibrillation, ischemic heart disease, estimated glomerular filtration rate (eGFR), hs-cTnT after dichotomizing into >≤14 ng/L, stroke etiology according to the TOAST classification, and both kinds of thrombolytic therapy to be ORFs of a poorer outcome in the whole stroke population. Including all significant ORFs into one multinomial logistic regression analysis model, only age, NIHSS score at entry, and arterial hypertension remained outcome-relevant in the total cohort. In a linear regression model with the same variables but adjusted for the NIHSS score at entry, both thrombolytic therapies did not exhibit any significance, indicating that the outcome is dependent only on the NIHSS score at entry.

#### 3.2.2. Relevance of the Predefined ORFs in the Outcome in Patients with Infratentorial and Supratentorial Strokes

The factors significantly related to the outcome in the univariate logistic regression analysis (Table 3) were age, diabetes mellitus, atrial fibrillation, ischemic heart disease, eGFR, and hs-cTnT in the infratentorial group, and age, the NIHSS score at entry, ischemic heart disease, eGFR, hs-cTnT, stroke etiology according to the TOAST classification, iv-lysis, and MT in the supratentorial group.

Serum levels of hs-cTnT are influenced by renal function. In our cohort (all patients), eGFR and hs-cTnT were correlated with each other (Spearman rho = −0.459, ≤0.0001), indicating that hs-cTnT increases with the decrease in eGFR. Therefore, we utilized numerous multinomial logistic regression analysis models with the dichotomized mRs outcome: in the infratentorial group, we tested a total of four models with age, diabetes mellitus, atrial fibrillation, and ischemic heart disease as the fixed variables. The additional variables were hs-cTnT and eGFR as either their continuous measures or their dichotomized measures (hs-cTnT ≤ 14 vs. > 14 ng/L, eGFR ≤ 60 vs. >60 mL/min/1.73 m^2^). In these multinomial models, hs-cTnT remained non-significantly related to the outcome in the infratentorial stroke group. In the supratentorial group, the fixed variables were ischemic heart disease, stroke etiology according to the TOAST classification, iv-lysis, and MT. The additional variables were hs-cTnT and eGFR in their dichotomized measures (hs-cTnT ≤ vs. >14 ng/L, eGFR ≤ 60 vs. >60 mL/min/1.73 m^2^) or their continuous measures (eGFR only). Both hs-cTnT and eGFR remained without relevance for the outcome of supratentorial stroke. If the dichotomized measures of eGFR and of hs-cTnT are combined as one score [0, both parameters were normal; 1, one parameter pathological (hs-cTnT > 14 or eGFR ≤ 60); or 2, both parameters pathological], this score exhibits a slightly higher AUC value than each of the two original measures, but the combined score is not superior to the original measures in multinomial analysis models. Finally, the variance inflation factor indicated that the dependency of age, eGFR, and hs-cTnT on each other was weak at best (VIF age/eGFR 1.33; age/hs-cTnT 1.10; and eGFR/hs-cTnT 2.13). A summary of the multinomial logistic regression analysis is presented in Table 4 and Table 5. In the supratentorial group, only age and NIHSS score at entry remained significantly related to outcome, while in the infratentorial group, atrial fibrillation and eGFR were significantly related to the outcome, and diabetes mellitus was marginally related due to the 95% CI range.

#### 3.2.3. Relevance of the Predefined ORFs in the Outcome in Patients with Supratentorial Strokes with or Without Mechanical Thrombectomy

The patients in the supratentorial stroke group without MT exhibited a mild stroke severity with a mostly very good outcome [NIHSS score at entry 2 (IQR 1–5), NIHSS score at 3 months 0 (IQR 0–1), mRs at entry 2 (IQR 1–3), and mRs 3 months 0 (IQR 0–1)]. The patients with an MT procedure were moderately to severely impaired [NIHSS score at entry 14 (IQR 8–18) and mRS at entry 4 (IQR 3–5] and had increased stroke severity by 3 months [NIHSS 3 (IQR 0–6.25) and mRs 3 (IQR 1–3)]; 47% of the 61 patients had a good outcome.

The factors significantly related to the outcome in the univariate logistic regression analysis (Table 6) were age in the patients without MT, NIHSS score at entry, atrial fibrillation, eGFR, hs-cTnT, and stroke origin according to the TOAST classification. In the patients with MT, ORFs were age, NIHSS score at entry, and arterial hypertension. MT patients without arterial hypertension had a good outcome 16 times and a poor outcome 10 times, while the MT patients with arterial hypertension exhibited a good outcome 12 times and a poor outcome 22 times (chi-squared test 4.07, *p* = 0.04; odds ratio 2.93, 95% CI 1.01–8.44).

In the patients without MT, we included age, NIHSS score at entry, hs-cTnT (as a continuous variable as well as the dichotomized variable alone or in combination with the dichotomized eGFR), and the dichotomized eGFR in the multinomial logistic regression analysis models. Only the NIHSS score at entry remained significantly related to a poor outcome (OR 1.52, 95% CI 1.31–1.72, *p* < 0.0001). In the group with MT, we included age, NIHSS score at entry, and arterial hypertension in the multinomial model; of the three variables, only NIHSS score at entry remained significantly related to poor outcomes (OR 1.22, 95% CI 1.08–1.39; *p* = 0.0007).

## 4. Discussion

The findings of the univariate logistic regression analysis for the total stroke population, the patients with infratentorial strokes, and those with supratentorial strokes are summarized in Figure 2. It can be observed that the number of ORFs decreases with the number of patients per group (see Limitations Section below). However, some points are relevant and should be addressed.

The classic vascular risk factors arterial hypertension and diabetes mellitus are significantly related to the outcome measure after 3 months if the total stroke population is considered. It is, therefore, important to stress that the treatment of these risk factors remains one of the focuses in early post-stroke care. Looking at the AUC results, we would like to suggest that the quality of post-stroke therapy in comprehensive stroke centers seems to be so effective that many risk factors nowadays provide at best only moderate predictive ability. One should therefore keep in mind that, as a consequence, an increasing number of patients is necessary to prove that these risk factors are still relevant.

In the supratentorial infarction groups, the clinical severity (NIHSS) of the stroke event is the most relevant ORF, while it is not predictive in the infratentorial infarction group. The main reason for this result could be that the NIHSS is not developed for infratentorial stroke. More likely, however, is that the patients with infratentorial strokes were significantly less severely impaired at entry and had a better chance of recovery than the patients with supratentorial strokes. Considering the multinomial analysis, it is surprising that chronic kidney disease, as indicated by an eGFR of ≤60 mL/min/1.73 m^2^, is an independent ORF for a poorer outcome in the infratentorial stroke group but not in the supratentorial one. Such a result has been reported previously [23], and our result underlines the relevance of chronic kidney disease for patients with cerebellar and brain stem infarctions. The same seems true for the association of infratentorial stroke with diabetes mellitus, which too has repeatedly been reported to increase the risk of a poorer outcome [24,25,26], although in our population the statistical significance is just marginal (reflecting some statistical weaknesses of the used tests and analysis models in this borderline area). In line with considerations that diabetes prefers the brain stem for establishing microangiopathy is that non-alcoholic fatty liver disease via increased insulin resistance is also associated with brain stem microangiopathy [27]. We emphasized this microcirculatory pathway because further microvascular damage occurs in the conditions of chronic kidney disease, which could explain why these two ORFs influenced stroke prognosis despite our result showing that the stroke origin according to the TOAST classification is of no relevance in the infratentorial stroke group. It is noteworthy but not explained by our data that among the cardiac diseases, atrial fibrillation but not ischemic heart disease is outcome-predictive in the infratentorial stroke group, while this relationship is inverse in the supratentorial stroke group. Finally, hyperlipidemia was identified to be associated with brain stem microangiopathy but not with supratentorial white matter lesions [28]. We could not determine the outcome relevance of hyperlipidemia in either of the vascular provinces, but the lack of outcome effect may also be caused by the fact that around 80% of our patients suffered from dyslipidemia. Another outcome marker established recently is the infarct size in both the supratentorial and the infratentorial stroke groups [as well as dissections of the intracranial arteries] [29,30,31].

In the supratentorial stroke group, MT improved the neurological deficit (as compared to the NIHSS score at entry) and the outcome after 3 months, a result in agreement with the previous groundbreaking studies [1,2,3,4,5]. The risk of a poorer outcome is increased if MT is applied and should not be misinterpreted in the sense that MT causes a poor outcome. Instead, the adjusted model indicates that the risk of a poorer outcome is simply increased because the initial neurological deficit is so severe that MT is indicated and recovery is less complete compared to the patients who were not in need of MT. Our results regarding iv-lysis should be interpreted in the same way. In the univariate analysis, arterial hypertension is associated with a poorer outcome in patients with MT. This is in line with reports on larger stroke patient populations with MT, which showed that higher (systolic) blood pressure levels during or within 24 h after the MT procedures are associated with a poorer outcome [32,33,34]; the clinical background could be that high blood pressure in patients with arterial hypertension is often more difficult to control for compared to patients without arterial hypertension in critical conditions. A known history of arterial hypertension may alert clinicians to blood pressure regulation difficulties in MT patients. That the stroke origin (TOAST) did not show outcome relevance in the patients with MT may be owed to the inhomogeneity of the stroke origin in the TOAST classification and the relatively small number of our MT patients.

In the univariate analysis, high-sensitivity cardiac Troponin-T was an ORF in all stroke groups apart from the MT patients. In the multinomial models, it dropped out of significance in all patient groups. Hs-cTnT serum levels depend on kidney function, and kidney function depends on age. However, our variance inflation factors indicate that the effects of collinearity between these variables are low. One could therefore conclude that the role of hs-cTnT as a stroke outcome predictor might be independent of age or kidney function but also that its value is still not clearly defined, apart from being a marker of cardiac diseases [10,11,12,13,14,15,16].

### Limitations

The major limitation of our study is the retrospective analysis and its observational characteristics. As a result, clear sample size estimation could not be performed a priori. Group sample size could therefore be a relevant issue as some parameters showed a wide 95% CI. To achieve a significant result at a power level of 0.8, the total sample size was calculated as 80 patients for diabetes mellitus as a risk factor for poor outcomes in the infratentorial group; 86 for arterial hypertension; 68 for atrial fibrillation; 64 for ischemic heart disease; 64 for dichotomized hs-cTnT; and 52 for dichotomized eGFR in the supratentorial group. To judge the role of hs-cTnT in this group, a sample size of 188 patients would have been necessary. Overall, our results are in agreement with previous observational retrospective cohort studies [23,24,25,26], thus relativizing the sample size considerations and supporting the hypothesis of Sigurdsson et al. [18] that ORFs may vary between strokes in different vascular territories. Our patients exhibited only mild to moderate impairment; therefore, the results may not be generalizable to a population of more severely affected stroke patients. Our performance of stroke center care [although based on the European Stroke Organisation guidelines (https://eso-stroke.org/guidelines/eso-guideline-directory/#acute-stroke, accessed on 1 January 2020)] may differ from those of other stroke centers, preventing our results from being generalized to all stroke centers. We also analyzed a cohort of patients who presented with a so-called temporal bone window for transcranial Doppler investigation; because the presence of such an ultrasound window depends on age, we may have analyzed a relatively young stroke population. Additionally, we cannot rule out that the dynamic cerebral autoregulation assessments influenced the clinical management of the patients with the result that blood pressure management was emphasized more carefully, with the possible consequence that blood pressure was less relevant as an ORF. Such a relationship would therefore have a remarkable effect that needs further exploration.

## 5. Conclusions

In our cohort of stroke patients, multinomial logistic regression analysis revealed that in a modern comprehensive stroke center care setting, the outcome of a supratentorial stroke depends primarily on the stroke severity upon admission, regardless of whether an MT was performed or not, and on the patient’s age. The modifiable vascular risk factors did not affect the 3-month outcome in this stroke subpopulation. The outcome of an infratentorial stroke depends on the presence of atrial fibrillation and chronic kidney disease, and marginally on diabetes mellitus. Our findings support the observations that depending on the affected vascular territory, different ORF profiles might be relevant for predicting the ischemic stroke outcome.

## Figures and Tables

**Figure 1 life-15-00633-f001:**
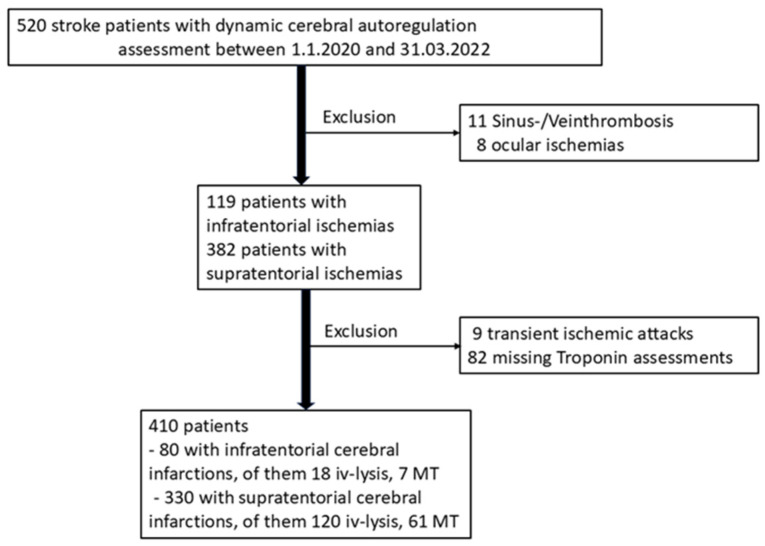
Flow chart of included patients. Iv-lysis indicates intravenous thrombolysis; MT indicates intra-arterial mechanical thrombectomy.

**Figure 2 life-15-00633-f002:**
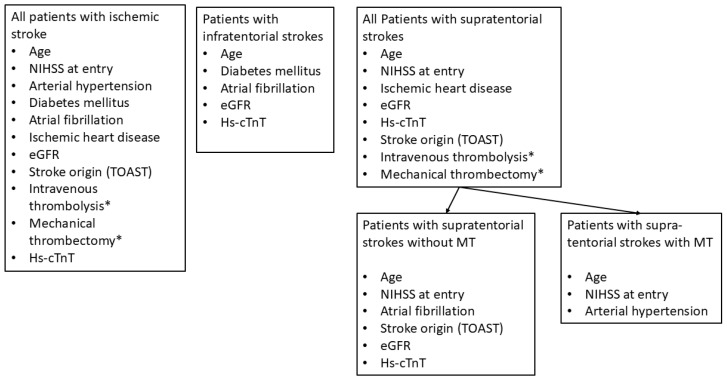
The outcome-relevant factors of poor outcome according to univariate logistic regression analysis in all patients and in different subgroups. eGFR, estimated glomerular filtration rate; Hs-cTnT, high-sensitivity cardiac Troponin T; NIHSS, National Institute of Health Stroke Scale; and TOAST, TOAST classification according to Reference [22]. * After adjusting for NIHSS score at entry, intravenous thrombolysis and mechanical thrombectomy were not significant anymore.

**Table 2 life-15-00633-t002:** Odds ratios of univariate logistic regression analysis and the receiver operating characteristic curve’s area under the curve analysis for the outcome-relevant factors and poor outcome (modified Rankin scale > 2, n = 70) vs. good outcome (modified Rankin scale < 2, n = 340) in all patients.

	Median (IQR) or Presence of (n) Odds Ratio	95% CI	*p*	ROCAUC
Age (years)	71 (59–58) 1.04	1.02–1.07	<0.0001	0.658
Sex (female/male)	115:295 1.53	0.89–2.26	0.11	0.546
NIHSS score at entry	3 (1–7) 1.22	1.17–1.29	<0.0001	0.857
Arterial hypertension	244 1.95	1.11–3.42	0.01	0.577
Diabetes mellitus	101 1.76	1.02–3.10	0.04	0.558
Metabolic syndrome	51 1.58	0.78–3.23	0.19	0.528
Dyslipidemia	320 1.02	0.54–1.91	0.94	0.502
Actual smoking	99 1.20	0.64–2.26	0.54	0.517
Body mass index (kg/m^2^)	25.8 (23.5–28.7) 1.01	0.95–1.07	0.60	0.522
Atrial fibrillation	65 2.13	1.15–3.98	0.01	0.559
Ischemic heart disease	79 2.45	1.39–4.48	0.001	0.581
Cardiac LVEF (%)	60 (55–64) 1.02	0.99–1.05	0.06	0.541
eGFR (mL/min/1.73 m^2^)	82 (66–90) 0.98	0.96–0.99	0.001	0.398
Hs-cTnT (ng/L)	11 (7–19) 0.99	0.99–1.01	0.35	0.668
TOAST classification of stroke	CE 121 LVD 81 Lac 75 Others 10 Unknown 123 0.74	0.61–0.88	0.0008	0.371
Intravenous thrombolysis	138 1.85	1.11–3.17	0.01	0.572
Mechanical thrombectomy	68 7.24	4.06–13.16	<0.0001	0.683
Hs-cTnT (ng/L) >14:≤14	145:265 2.77	1.64–4.69	<0.0001	0.622
eGFR (mL/min/1.73 m^2^) ≤60:>60	75:335 2.69	1.51–4.84	<0.0001	0.587
Hs-cTnT (ng/L) >14 vs. ≤14 Combined with eGFR (mL/min/1.73 m^2^) ≤60 vs. >60	Both normal 242 One of the two pathological 116 Both pathological 52 2.11	1.51–2.99	<0.0001	0.635

eGFR, estimated glomerular filtration rate; Hs-cTnT, high-sensitivity cardiac Troponin T; LVEF, left-ventricular ejection fraction; NIHSS, National Institute of Health Stroke Scale; odds ratio, odds ratio to be allocated in the mRs > 2 group; 95% CI, 95% confidence interval; *p*, level of significance; ROCAUC, receiver operating characteristic curve’s area under the curve; and TOAST, TOAST classification according to Reference [22] (CE, cardio-embolism; LVD, large vessel disease; Lac, lacunar; Others, other defined causes of stroke; and Unknown, no cause found or multiple causes). Hs-cTnT > 14 ng/L indicates a Troponin value above the 95% CI of the test’s upper limit range; eGFR ≤60 indicates chronic kidney disease.

**Table 3 life-15-00633-t003:** Odds ratios of univariate logistic regression analysis and the receiver operating characteristic curve’s area under the curve analysis for the outcome-relevant factors and poor outcome (modified Rankin scale > 2)_ vs. good outcome (modified Rankin scale < 2) in the cohort of patients with either infratentorial or supratentorial stroke. The outcome was poor in 10 patients in the infratentorial stroke group and in 60 patients in the supratentorial group.

			Infratentorial Stroke (n = 80)			Supratentorial Stroke (n = 330)		
	Median (IQR) or Presence of (n) Odds Ratio	95% CI	*p*	ROCAUC	Median (IQR) or Presence of (n) Odds Ratio	95% CI	*p*	ROCAUC
Age (years)	70 (58–76) 1.07	1.01–1.15	0.01	0.696	71 (59–78) 1.04	1.01–1.06	<0.0001	0.646
Sex (female/male)	15:65 2.05	0.45–9.39	0.35	0.564	100:230 1.41	0.79–2.56	0.23	0.538
NIHSS score at entry	2 (1–4) 0.98	0.84–1.12	0.69	0.545	3 (1–7) 1.31	1.23–1.40	<0.0001	0.913
Arterial hypertension	45 4.66	0.92–23.54	0.06	0.679	199 1.66	0.91–3.07	0.08	0.559
Diabetes mellitus	24 4.30	1.07–17.49	0.03	0.671	77 1.52	0.82–2.86	0.18	0.540
Metabolic syndrome	9 2.24	0.38–13.11	0.38	0.550	42 1.47	0.68–3.23	0.32	0.524
Dyslipidemia	62 1.17	0.22–6.31	0.83	0.514	258 0.99	0.50–1.96	0.52	0.499
Actual smoking	18 0.63	0.14–2.82	0.54	0.542	81 1.36	0.69–2.77	0.35	0.471
Body mass index (kg/m^2^)	25.9 (24.2–28.4) 1.04	0.86–1.15	0.66	0.516	27.5 (23.1–28.7) 0.98	0.91–1.04	0.50	0.476
Atrial fibrillation	8 10.91	2.12–56.92	0.004	0.671	57 1.58	0.80–3.15	0.18	0.536
Ischemic heart disease	9 8.58	1.78–42.19	0.006	0.664	70 1.97	1.06–3.72	0.03	0.563
Cardiac LVEF (%)	60 (55–64) 0.94	0.86–1.01	0.09	0.350	60 (55–64) 0.99	0.95–1.00	0.20	0.479
eGFR (mL/min/1.73 m^2^)	83 (68–91) 0.96	0.92–0.98	0.001	0.207	81 (66–90) 0.990	0.96–0.999	0.05	0.438
Hs-cTnT (ng/L)	10 (8–16) 1.03	1.01–1.06	0.004	0.795	11 (6–20) 1.00	0.99–1.001	0.45	0.644
TOAST classification of stroke	CE 15 LVD 28 Lac 18 Others 1 Unknown 18 0.69	0.39–1.23	0.17	0.376	106 53 57 9 105 0.74	0.61–0.89	0.001	0.373
Intravenous thrombolysis	18				120			
	All with good outcome			0.371	2.22	1.33–4.17	0.002	0.603
Mechanical thrombectomy	7				61			
	All with good outcome			0.450	9.77	5.17–18.73	<0.0001	0.722
Hs-cTnT (ng/L) >14:≤14	23:57 7.84	1.78–34.80	0.005	0.735	122:208 2.27	1.29–4.03	0.004	0.600
eGFR (mL/min/1.73 m^2^) ≤60:>60	15:65 10.07	2.34–44.15	0.001	0.735	60:270 2.07	1.08–4.00	0.03	0.562
Hs-cTnT (ng/L) >14 vs. ≤14 Combined with eGFR (ml/min/1.73 m_2_) ≤60 vs. >60	5.58	Both normal 51 One pathological 20 Both pathological 9 2.04–15.18	0.0006	0.802	191 96 43 1.80	1.23–2.60	0.001	0.603

eGFR, estimated glomerular filtration rate; Hs-cTnT, high-sensitivity cardiac Troponin T; IQR, interquartile range; LVEF, left-ventricular ejection fraction; NIHSS, National Institute of Health Stroke Scale; odds ratio, odds ratio to be allocated in the mRs > 2 group; 95% CI, 95% confidence interval; *p*, level of significance; ROCAUC, receiver operating characteristic curve’s area under the curve; and TOAST, TOAST classification according to Reference [22] (CE, cardio-embolism; LVD, large vessel disease; Lac, lacunar; Others, other defined causes of stroke; and Unknown, no cause found or multiple causes). Hs-cTnT > 14 ng/L indicates a Troponin value above the 95% CI of the test’s upper limit range; eGFR ≤60 indicates chronic kidney disease.

**Table 4 life-15-00633-t004:** Multinomial logistic regression analysis of the outcome-relevant factors significant in the univariate analysis for predicting an outcome of mRs > 2 in the infratentorial stroke group.

	Odds Ratio	95% CI	*p*
Age	1.01	0.94–1.09	0.65
Diabetes mellitus	7.69	0.96–62.63	0.05
Atrial fibrillation	13.73	1.05–181.89	0.04
Ischemic heart disease	9.97	0.57–169.54	0.10
Hs-cTnT (ng/L)	1.01	0.97–1.04	0.46
eGFR (mL/min/1.73 m^2^)	0.96	0.91–0.99	0.02
Hs-cTnT (ng/L) >14 vs. ≤14 combined with eGFR (mL/min/1.73 m^2^) ≤60 vs. >60	4.05	1.17–14.05	0.02

CI, confidence interval; eGFR, estimated glomerular filtration rate; Hs-cTnT, high-sensitivity cardiac Troponin T; and *p*, level of significance.

**Table 5 life-15-00633-t005:** Multinomial logistic regression analysis of the outcome-relevant factors significant in the univariate analysis for predicting an outcome of mRs > 2 in the supratentorial stroke group.

	Odds Ratio	95% CI	*p*
Age	1.04	1.01–1.08	0.01
NIHSS score at entry	1.32	1.22–1.44	<0.0001
Ischemic heart disease	1.66	0.66–4.18	0.27
TOAST	0.99	0.77–1.26	0.93
Intravenous thrombolysis	0.59	0.25–1.38	0.22
Mechanical thrombectomy	1.50	0.95–2.41	0.07
Hs-cTnT (ng/L) >14 vs. ≤14	1.10	0.44–2.71	0.83
eGFR (mL/min/1.73m^2^) ≤60 vs. >60	0.81	0.28–2.24	0.67
Hs-cTnT (ng/L) >14 vs. ≤14 combined with eGFR (mL/min/1.73 m^2^) ≤60 vs. >60	0.96	0.52–1.72	0.88

CI, confidence interval; eGFR, estimated glomerular filtration rate; Hs-cTnT, high-sensitivity cardiac Troponin T; NIHSS, National Institute of Health Stroke Scale; TOAST, TOAST classification according to Reference [22]; and *p*, level of significance.

**Table 6 life-15-00633-t006:** Odds ratios of univariate logistic regression analysis and the receiver operating characteristic curve’s area under the curve analysis for the outcome-relevant factors and poor outcome (modified Rankin scale > 2) vs. good outcome (modified Rankin scale < 2) in the cohort of patients with supratentorial infarctions with and without mechanical thrombectomy (MT) procedures. A poor outcome was observed in 28 patients in the group without MT and in 32 patients in the group with MT.

	Supratentorial Stroke Without MT (n = 269)				Supratentorial Stroke with MT (n = 61)			
	Median (IQR) or Presence of (n) Odds Ratio	95% CI	*p*	ROCAUC	Median (IQR) or Presence of (n) Odds Ratio	95% CI	*p*	ROCAUC
Age	71 (59–78) 1.03	1.00–1.07	0.02	0.629	73 (61–81) 1.05	1.00–1.09	0.02	0.675
Sex (female/male)	82:187 2.15	0.97–4.79	0.06	0.589	17:44 0.98	0.31–3.18	0.96	0.497
NIHSS score at entry per scale point	2 (1–5) 1.51	1.33–1.72	<0.0001	0.915	14 (8–18) 1.20	1.07–1.34	0.0006	0.786
Arterial hypertension	165 1.64	0.69–3.92	0.25	0.556	40 2.91	1.00–8.64	0.04	0.629
Diabetes mellitus	61 1.15	0.46–2.87	0.75	0.513	15 3.12	0.84–11.68	0.08	0.600
Metabolic syndrome	33 1.20	0.39–3.78	0.73	0.5113	9 1.91	0.41–8.81	0.37	0.540
Dyslipidemia	213 0.94	0.36–2.45	0.90	0.506	44 1.68	0.52–5.49	0.37	0.551
Actual smoking	70 0.93	0.37–2.30	0.49	0.507	11 0.68	0.17–2.59	0.56	0.471
Body mass index (kg/m^2^)	25.7 (23–28.7) 0.96	0.87–1.04	0.34	0.446	25.0 (24.0–28.7) 1.01	0.90–1.14	0.74	0.511
Atrial fibrillation	44 2.74	1.15–6.61	0.02	0.587	13 0.46	0.12–1.67	0.23	0.435
Ischemic heart disease	57 1.95	0.82–4.61	0.12	0.563	13 3.78	0.89–16.01	0.06	0.602
Cardiac LVEF (%)	60 (55–63) 0.97	0.92–1.00	0.10	0.337	60 (51–65) 1.01	0.96–1.05	0.56	0.532
eGFR (mL/min/1.73 m^2^)	81 (67–90) 0.99	0.95–1.00	0.09	0.411	78 (58–90) 1.00	0.97–1.02	0.94	0.518
Hs-cTnT (ng/L)	11 (6–19) 1.003	1.0002–1.006	0.02	0.633	14 (7–26) 1.00	0.98–1.00	0.47	0.591
TOAST classification of stroke	CE 76 LVD 37 Lac 57 Others 3 Unknown 96 0.76	0.58–0.98	0.03	0.380	29 16 7 0 9 0.96	0.67–1.36	0.81	0.501
Intravenous thrombolysis	81 1.85	0.83–4.17	0.12	0.571	0.93	38 0.31–2.71	0.88	0.491
Hs-cTnT (ng/L) >14:≤14	92:117 2.43	1.11–5.43	0.02	0.608	1.29	30:31 0.46–3.69	0.60	0.533
eGFR (mL/min/1.73 m^2^) ≤60:>60	45:224 2.69	1.12–6.45	0.03	0.586	1.00	16:45 0.30–3.30	0.99	0.500
Hs-cTnT (ng/L) >14 vs. ≤14 Combined with eGFR (mL/min/1.73 m^2^) ≤60 vs. >60	Both normal 162 One pathological 77 Both pathological 30 2.03	1.22–3.42	0.006	0.624	1.10	28 20 13 0.57–2.15	0.74	0.517

eGFR, estimated glomerular filtration rate; Hs-cTnT, high-sensitivity cardiac Troponin T; LVEF, left-ventricular ejection fraction; NIHSS, National Institute of Health Stroke Scale; OR, odds ratio to be allocated in the mRs >2 group; 95% CI, 95% confidence interval; *p*, level of significance; ROCAUC, receiver operating characteristic curve’s area under the curve; and TOAST, TOAST classification according to Reference [22] (CE, cardio-embolism; LVD, large vessel disease; Lac, lacunar; Others, other defined causes of stroke; and Unknown, no cause found or multiple causes). Hs-cTnT > 14 ng/L indicates a Troponin value above the 95% CI of the test’s upper limit range; eGFR ≤ 60 indicates chronic kidney disease.

## Data Availability

The corresponding author can provide all data for this study upon reasonable request.

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
