# Peer review of "Outcome Predictor Differences in Infratentorial and Supratentorial Ischemic Stroke"

_life, 2025, doi:10.3390/life15040633_

Round 1
Reviewer 1 Report
Comments and Suggestions for Authors
Retrospective analysis of prognostic factors in stroke patients.
The authors statistically assessed the risk factor in the group:
1/ all patients (n=410),
2/ patients with infratentorial infarctions only (n=80),
3/ all patients with supratentorial infractions (n=330),
4/ patients with supratentorial infarctions without (n=270)
5/ with mechanical thrombectomy (n=60).
Outcome was classified by modified Rankin scale ≤2 (good) vs >2 (poor) after three months.
In the conclusions, the authors stated that poor outcome was associated with:
group 1/ - age, stroke severity (NIHSS at hospital entry), ischemic heart disease, eGFR, hs-cTnT, stroke origin, intravenous thrombolysis, and mechanical thrombectomy (MT);
group 2/ - atrial fibrillation, eGFR and diabetes mellitus; group 3/ NIHSS and age
group 4/ age, NIHSS, atrial fibrillation, stroke origin, eGFR and hs-cTnT.
group 5/ age, NIHSS, and arterial hypertension.
So regardless of location, the main factor of poor prognosis was: stroke severity (NIHSS), age, presence of FA and diabetes. And these are previously known factors and are repeated in this analysis.
Notes:
1/ how should mechanical thrombectomy or thrombolysis be understood in group 1 among poor prognostic factors?
2/ what was the range of stroke severity - did I understand correctly from the text that the maximum value was 7 in the supratentorial group, and the mean/median (?) was 2 and 3, i.e. strokes of very low severity - so what were the indications for qualification for thrombolysis or technical thrombectomy?
3/ why may troponin be a risk factor in the group of patients without technical thrombectomy but not in the group where treatment was applied?
Author Response
Dear reviewer,
many thanks for your helpful comments to improve the manuscript. We hope our responses and changes in the manuscript are in your intention. We refer to the revised manuscript in our responses, please use the clean manuscript life-3481388 manuscript v2 english-edited rev1.
1/ how should mechanical thrombectomy or thrombolysis be understood in group 1 among poor prognostic factors? Response: We understand your comment completely. As reommended by the other reviewers, we adjusted each of the thrombolytic therapies to the NIHSS score at entry with the result that both thrombolytic therapies were without significance anymore (lines 192-195). We marked this result in figure 2 by a * and explained it in the figure legend (lines 324-325). We further warned in the discussion that our result should not be misinterpreted (368-372): "In the supratentorial stroke group, MT improved the neurological deficit (as compared to the NIHSS at entry) and the outcome after 3 months, a result in agreement with the previous ground breaking studies [1-5]. That the risk of poorer outcome is increased if MT is applied should not be misinterpreted in the sense that MT causes the poor outcome. Instead, the risk of poorer outcome is simply increased because the initial neurological deficit is so severe that MT is indicated, and recovery less complete compared to the patients who were not in need of MT. Our results of the intravenous thrombolysis should be interpreted in the same way."
2/ what was the range of stroke severity - did I understand correctly from the text that the maximum value was 7 in the supratentorial group, and the mean/median (?) was 2 and 3, i.e. strokes of very low severity - so what were the indications for qualification for thrombolysis or technical thrombectomy? Response: Stroke severity was presented in the paragraph "Relevance of the predefined risk factors in the outcome in patients with supratentorial strokes with or without mechanical thrombectomy." lines 270-284. To get information on stroke severity earlier in the manuscript we included the range of stroke severity for the infra- and the supratentorial stroke groups into table 1.
3/ why may troponin be a risk factor in the group of patients without technical thrombectomy but not in the group where treatment was applied? Response: hs-cTnT was a risk factor in the univariate analysis over all patients, in the patients with infratentorial strokes and over all supratentorial strokes. We think that the thrombectomy patient group is mixed up with to many different stroke etiologies and the number of this group is too low to compensate for this mixture. We discussed at least the group size as one reason that hs-cTnT lacks significance in the thrombectomy group (Lines 384 - 391).
Reviewer 2 Report
Comments and Suggestions for Authors
The authors explored what factors predicted chronic poor outcome in patients with infratentorial or supratentorial ischemic stroke. Multivariable logictic regression analysis revealed that AF and eGFR values were independent predictors of poor functional outcome in patients with infratentorial ischemic stroke. However, no predictors were detected in patients with supratentorial ischemic stroke.
Here, some problems can be raised in the manuscript, the authors need to address them shown below.
1. Basically, outcome predictors should be modifiable. The authors indicate that only NIHSS remained associated with poor outcome in patients with supratentorial ischemic stroke. However, NIHSS on admission is not a modifiable factor, and it is quite natural that worse NIHSS leads to poorer functional outcome in chronic phase. Therefore, the authors should conclude that no outcome predictors sis not emerge in patients with supratentorial ischemic stroke not to mislead readers.
2. In Abstract, the results in univariable logistic regression analysis are should be deleted. Instead, the authors have to describe methods in detail. The methods are lacking.
3. In Table 4, ranges of 95% CI in AF and ischemic heart disease are too wide. These may be because the number of patients with AF or ischemic heart disease are only 8 or 9, respectively. That is, the numbers are too small. In the case, exact logistic regression model is often used.
Author Response
Dear reviewer,
many thanks for your helpful comments to improve the manuscript. We hope our responses and changes in the manuscript are in your intention. We refer to the revised manuscript in our responses, please use the clean manuscript life-3481388 manuscript v2 english-edited rev1
- Basically, outcome predictors should be modifiable. The authors indicate that only NIHSS remained associated with poor outcome in patients with supratentorial ischemic stroke. However, NIHSS on admission is not a modifiable factor, and it is quite natural that worse NIHSS leads to poorer functional outcome in chronic phase. Therefore, the authors should conclude that no outcome predictors sis not emerge in patients with supratentorial ischemic stroke not to mislead readers. Respone. We agree to your suggestion but we want to kep in mind that commonly age and NIHSS are considered "risk" factors of a poor outcome. That NIHSS is still a risk factor despite the invention of IV-lysis and mechanical thrombectomy illustrates that there is still a long way to go in stroke care to overcome this factor. Not to be misunderstood, we hope to have clarified your consideration in the discussion section, Figure 2 (legend) and lines 192-194 and 366-372.
- In Abstract, the results in univariable logistic regression analysis are should be deleted. Instead, the authors have to describe methods in detail. The methods are lacking. Response: We have re-written the abstract by dropping the univariate analysis, and by including more of methology. We hope we have met your considerations accordingly (Abstract, page 2).
- In Table 4, ranges of 95% CI in AF and ischemic heart disease are too wide. These may be because the number of patients with AF or ischemic heart disease are only 8 or 9, respectively. That is, the numbers are too small. In the case, exact logistic regression model is often used. Response: Thank you for this consideration which is in line with one consideration of reviewer 4 who suggested to look after the sample size if the 95%CI is that wide. There were some other factors which also had had a wide 95%CI. We performed the sample size analysis and hope you can agree to that approach instead of using the exact logistic regression analysis approach which is also not without problems. Please, see the statistic section (lines 136 - 137, and discussion section, lines 396-401).
Reviewer 3 Report
Comments and Suggestions for Authors
Dear sir,
I have the following comments regarding your article
Line 46- needs correction
Introduction
Needs to introduce existing supra and infratentorial infarction, including thrombolysis and thrombectomy benefit. The need of this study and its hypothesis needs clarification
Statistical analysis
The different groups of patients could have been done by multiple comparison for outcome
Results:
Flow chart may mention number of patients with and without thrombolysis/thrombectomy
Table 2 should mention number of patients with poor & good outcome for clinical relevance
Table 3. Is a statistical table, make it clinically relevant
Overall, the results need to reformat to be clinically relevant. The included parameters should have a justification based on the hypothesis.
Discussion
Started with Fig 2? Is it the norma of the journal?
Needs redrafting
Author Response
Dear reviewer,
many thanks for your helpful comments to improve the manuscript. We hope our responses and changes in the manuscript are in your intention. We refer to the revised manuscript in our responses, please use the clean manuscript life-3481388 manuscript v2 english-edited rev1.
Comment: Line 46- needs correction. Response: mayor is changed to major (line 50)
Comment: Needs to introduce existing supra and infratentorial infarction, including thrombolysis and thrombectomy benefit. The need of this study and its hypothesis needs clarification. Response: the introduction section was re-written to include the benefits (page 2, paragraph 1) , and we hope to have clarified the purpose of the study (page 2, paragraph 2)
Comment: The different groups of patients could have been done by multiple comparison for outcome. Response: We did this analysis using Kruskal-Wallis-test. The result were the same as outlined in the manuscript that the patients with MT had had a worser outcome than all others. We were interested in the outcome determining factors in each group and not on the outcome between the groups. We, therfore, don't think the reporting of the outcome analysis between the different stroke groups is necessary for our intention.
Comment: Flow chart may mention number of patients with and without thrombolysis/thrombectomy. Response: Your fine recommendation is now included into the flow chart. See Figure 1
Comment: Table 2 should mention number of patients with poor & good outcome for clinical relevance. Response: We layed out the tables 2,3 and 6 in the same manner to include the clinical data. The outcome data are included into the heading of tables. We additionally included into the tables for each factor either the median over the whole population or the number of patients for example suffering from arterial hypertension. We hope the tables are now more useful from the clinical viewpoint.
Comment: Overall, the results need to reformat to be clinically relevant. The included parameters should have a justification based on the hypothesis. Response: The reason/justification which outcome relevant factors should be included into our analysis is provided in the introduction section (page 2, paragraph 1). We hope, the outlining meets your comment. The clinical relevance of our results is included into tables 2,3, and 6 to allow for the results to be delineated from these tables.
Started with Fig 2? Is it the norma of the journal? Response We changed the wording of this sentence and hope we met your consideration.
Comment: Discussion needs redrafting. Response: Due to the changes made because of all reviewer's comments, large parts of the discussion have been redrafted.
Reviewer 4 Report
Comments and Suggestions for Authors
This retrospective cohort study examines differential outcome predictors between infratentorial and supratentorial ischemic strokes, identifying distinct risk profiles (e.g., atrial fibrillation and renal dysfunction in infratentorial vs. NIHSS and age in supratentorial strokes) under modern stroke care. While reinforcing known predictors like NIHSS, it highlights underrecognized associations, such as chronic kidney disease in posterior circulation strokes. Recent studies emphasize territorial-specific pathophysiology (e.g., posterior circulation stroke mechanisms) and biomarker utility (e.g., hs-cTnT) in stroke prognosis. This work aligns with emerging interest in individualized risk stratification but uniquely contrasts infra-/supratentorial predictors, a niche less explored compared to anterior/posterior circulation comparisons.
- The infratentorial cohort (n=80) risks underpowered analyses (e.g., atrial fibrillation OR=13.73 with wide CI: 1.05–181.89; Hs-cTnT with 1.17-14.05). A priori power calculation or sensitivity analysis is needed to validate reliability.
- Acknowledge NIHSS’s bias toward supratentorial deficits. Consider supplementing with posterior circulation-specific scales for infratentorial strokes.
- MT patients had higher baseline NIHSS (median 14 vs. 2). Clarify whether multivariate models adjusted for baseline severity when associating MT with poor outcome.
- hs-cTnT’s loss of significance in multivariate models warrants discussion—is it a mediator (via renal/cardiac pathways) or redundant to eGFR/age?
- Report variance inflation factors (VIFs) for variables like age/eGFR in regression models to exclude collinearity.
- 6. mRS at 3 months was clinician-assessed. Specify blinding status to avoid observer bias.
- It is difficult to be certain that the P-value for Diabetes mellitus in Table 4 is 0.05, given that the confidence interval is 0.96–62.63. Please verify the statistical results and check whether Diabetes mellitus is an independent prognostic factor after multivariable logistic regression in the full text (including the abstract, results, discussion, and conclusion sections).
- Update the references. Although the authors have cited many studies, there is insufficient discussion of references related to both supratentorial and infratentorial aspects.
- Maintain consistency in the number of decimal places for P-values as much as possible. Please change P = 0.0000 to P < 0.001.
- Pay attention to the placement of figure captions, which are generally positioned below the figures.
- What does the last sentence mean? “This section is not mandatory but can be added to the manuscript if the discussion is unusually long or complex.”
- Multiple mechanical thrombectomy(MT)etc. abbreviation. Please check the entire manuscript carefully. If the attitude is not serious, I will reject the paper.
Final Recommendation: Major Revision (Address statistical power, clarify methods, and refine language).
Comments on the Quality of English LanguageCan be improved.
Author Response
Dear reviewer,
many thanks for your helpful comments to improve the manuscript. We hope our responses and changes in the manuscript are in your intention. We refer to the revised manuscript in our responses, please use the clean manuscript life-3481388 manuscript v2 english-edited rev1. The English was edited so we hope thectext improved its accuracy.
Comment: The infratentorial cohort (n=80) risks underpowered analyses (e.g., atrial fibrillation OR=13.73 with wide CI: 1.05–181.89; Hs-cTnT with 1.17-14.05). A priori power calculation or sensitivity analysis is needed to validate reliability. Response: We are aware of this risk and discussed it in the discussion section. Due to the observational study design over a defined time period a power calculation prior to study start was not possible. We tried to reflect on this topic by providing some group size calculations in the discussion section (lines 397 - 402).
Comment: Acknowledge NIHSS’s bias toward supratentorial deficits. Consider supplementing with posterior circulation-specific scales for infratentorial strokes. Response: In our stroke center we monitor all patients by means of NIHSS as that is the requirement of the law. We did not use a special stroke scale for infratentorial strokes. There are a few stroke scales for infratentorial strokes but none has successfully been proven to be useful in clinical practice. To score the patients retrospectively is not a valid approach as symptomatology of infratentorial can rapidly change and provides therefore a hugh potential for misclassifications. As a result, unfortunately we can't provide data with a valid infratentorial stroke scale. Your comment is correct and shows some of the weaknesses of current stroke care.
Comment: mRS at 3 months was clinician-assessed. Specify blinding status to avoid observer bias. Response: We included into the Method sectio Seeting the follwing sentence "the clinical scoring three months after the ischemic event was performed by the personal of the outpatient service team which was not involved in the scoring at hospital entry and at the stroke unit" (lines 105-107)
Comment: MT patients had higher baseline NIHSS (median 14 vs. 2). Clarify whether multivariate models adjusted for baseline severity when associating MT with poor outcome. Response: Because thrombolytic therapies can be applied on every stroke patient and is, therefore, relevant for the total stroke population, we included in the section "Relevance of the predefined risk factors in the outcome in the total stroke population" the result of your recommended analysis. "In a linear regression multivariate model with the same variables but adjustd for NIHSS at entry, both thrombolytic therapies lost their significance, indicating that outcome is not dependent on whether MT or iv-lysis was performed or not (lines 192-194). It supports the result of the multinomial analysis (table 5) in which MT lost also its significance. In figure 2 we marked intravenous thrombolysis and mechanical thrombectomy with a * and stated in the figure legend the result of this analysis
Comment: It is difficult to be certain that the P-value for Diabetes mellitus in Table 4 is 0.05, given that the confidence interval is 0.96–62.63. Please verify the statistical results and check whether Diabetes mellitus is an independent prognostic factor after multivariable logistic regression in the full text (including the abstract, results, discussion, and conclusion sections). Response: The results reported are the results of a linear logistic regression analysis with several input-variables (table 4) and one output-variable (dichotomized mRs), termed multinomial logistic regression analysis. If the same (multiple) input-variables and the (single) outcome variable are included in a not logit transformed "normal" linear regression model, Diabetes mellitus exhibits a significance level of p=.03 with an odds ratio of 1.15 (95%CI 1.01-1.31). It seems that in this borderzone area around .05 the statistical analysis varies depending on the tests and the models used. To admit such uncertainties we stated throughout the manuscript that the influence of diabetes is (just) marginal. lines 28/29, 254-256, 347-349
Commment: Report variance inflation factors (VIFs) for variables like age/eGFR in regression models to exclude collinearity. hs-cTnT’s loss of significance in multivariate models warrants discussion—is it a mediator (via renal/cardiac pathways) or redundant to eGFR/age? Response: Please allow us to responsd to both your suggestions together as they regard the same topic. We report the VIFs on lines 250-252 by stating "Finally, the variance inflation factors indicated that the dependency of age, eGFR and hs-cTnT on each other were weak at best (VIF age/eGFR 1.33; age/hs-cTnT 1.10; eGFR/hs-cTnT 2.13).". As a consequence, we admit in the discussion section (lines 384-390): In the univariate analysis, high-sensitive cardiac Troponin-T was an outcome risk factor in all stroke groups apart from the MT patients. In the multinomial models, it dropped out of significance in all patient groups. Hs-cTnT serum levels depend on kidney function, and kidney function depend on age. However, the reported variance inflation factors indicate that effects of collinearity between these variables is low.. One could therefore conclude that the role of hs-cTnT as a stroke outcome predictor might be independent of age or kidney function, but that ist value is still not clearly defined yet, apart from being a marker of cardiac diseases [10-16].
Comment: Update the references. Although the authors have cited many studies, there is insufficient discussion of references related to both supratentorial and infratentorial aspects. Response: We hope to have actualized the references as you intended. References 27-31 are the new onces.
Comment: Maintain consistency in the number of decimal places for P-values as much as possible. Please change P = 0.0000 to P < 0.001. Response: We changed as recommended throughout the manuscript.
Comment: Pay attention to the placement of figure captions, which are generally positioned below the figures. Response: We changed the positioning of the captations. Thank you for this hint.
Comment: What does the last sentence mean? “This section is not mandatory but can be added to the manuscript if the discussion is unusually long or complex.” Response: This is a sentence in the template which we missed to delete. It has no meaning for the manuscript. Sorry.
Comment: Multiple mechanical thrombectomy (MT) etc. abbreviation. Please check the entire manuscript carefully. If the attitude is not serious, I will reject the paper. Response: Thank's for this comment, we hope this is now no issue anymore.
Round 2
Reviewer 2 Report
Comments and Suggestions for Authors
- Basically, outcome predictors should be modifiable. The authors indicate that only NIHSS remained associated with poor outcome in patients with supratentorial ischemic stroke. However, NIHSS on admission is not a modifiable factor, and it is quite natural that worse NIHSS leads to poorer functional outcome in chronic phase. Therefore, the authors should conclude that no outcome predictors did not emerge in patients with supratentorial ischemic stroke not to mislead readers.
2. In Table 4, ranges of 95% CI in AF and ischemic heart disease are too wide. These may be because the number of patients with AF or ischemic heart disease are only 8 or 9, respectively. That is, the numbers are too small. In the case, exact logistic regression model is often used.
Author Response
Dear reviewer,
again many thanks for your helpful comments !
Comment 1. Basically, outcome predictors should be modifiable. The authors indicate that only NIHSS remained associated with poor outcome in patients with supratentorial ischemic stroke. However, NIHSS on admission is not a modifiable factor, and it is quite natural that worse NIHSS leads to poorer functional outcome in chronic phase. Therefore, the authors should conclude that no outcome predictors did not emerge in patients with supratentorial ischemic stroke not to mislead readers. Response: We hope to be concise now by inventing the term "outcome-relevant factors (ORF)" which includes all factors shown to be relevant for outcome; that includes the vascular risk factors but also age and NIHSS (e.g.). The modifiable vascular risk factors as BP, DM, hyperlipidemia are always referred to "vascular risk factors". So we used two different terms throughout the manuscript and hope to have met yout intention accordingly.
Comment 2. In Table 4, ranges of 95% CI in AF and ischemic heart disease are too wide. These may be because the number of patients with AF or ischemic heart disease are only 8 or 9, respectively. That is, the numbers are too small. In the case, exact logistic regression model is often used. Response: We agree that for small samples sizes exact logistic regression analysis is a better way to analyze binomial data such as DM or atrial fibrillation in their presentation yes or no. After speaking with our statisticians their advise was: because exact logistic regression is not a good option for continuous variables such as age or eGFR which are also in our regression model; they advised us to stay with the multinomial logistic regression model and have to accept the wide 95% CIs. We hope you can accept our considerations.
Reviewer 3 Report
Comments and Suggestions for Authors
Dear sir,
No further comment. May publishAuthor Response
Many thanks for your kind reply !
Reviewer 4 Report
Comments and Suggestions for Authors
The manuscript exhibits minor issues not fully addressed. For example, there are small typographical errors such as “supratentorial infractions” instead of “infarctions” and the use of a comma instead of a decimal point in a numerical value (e.g., “9,77” should be “9.77” in Table 4). These inconsistencies affect the professionalism and clarity of the presentation. The authors are urged to check the entire manuscript for uniform terminology and correct formatting. Ensuring that all abbreviations are used consistently (with a single definition) and that all numerical and spelling errors are corrected will improve the paper’s readability and demonstrate the necessary attention to detail.
Comments on the Quality of English LanguageShould be improved.
Author Response
Dear Reviewer,
many thanks again for your kind comments !
Comment: The manuscript exhibits minor issues not fully addressed. For example, there are small typographical errors such as “supratentorial infractions” instead of “infarctions” and the use of a comma instead of a decimal point in a numerical value (e.g., “9,77” should be “9.77” in Table 4). These inconsistencies affect the professionalism and clarity of the presentation. The authors are urged to check the entire manuscript for uniform terminology and correct formatting. Ensuring that all abbreviations are used consistently (with a single definition) and that all numerical and spelling errors are corrected will improve the paper’s readability and demonstrate the necessary attention to detail. Response: As you can see in the revised manuscript (please look at the pdf file) we changed several grammatical and speeling errors and hope the manuscript will find your approval now.